# Evaluation of the Effects of Autologous Leukocyte- and Platelet-Rich Fibrin Membranes for Treating Chronic Wounds: A Prospective Study [note 1]

**DOI:** 10.3390/ani15010112

**Published:** 2025-01-06

**Authors:** Federica Aragosa, Gerardo Fatone, Chiara Caterino, Stefano Cavalli, Alfonso Piscitelli, Rosario Vallefuoco, Francesco Lamagna, Giovanni Della Valle

**Affiliations:** 1Department of Veterinary Medicine and Animal Production, University of Naples “Federico II”, 80137 Naples, Italy; federica.aragosa@unina.it (F.A.); fatone@unina.it (G.F.); stefano.cavalli@unina.it (S.C.); lamagna@unina.it (F.L.); giovanni.dellavalle@unina.it (G.D.V.); 2Department of Agricultural Sciences, University of Naples “Federico II”, 80055 Portici, Italy; alfonso.piscitelli@unina.it; 3Pride Veterinary Referrals, Derby DE24 8HX, UK; rosario.vallefuoco@scarsdalevets.com

**Keywords:** platelet concentrates, L-PRF, second intention, spontaneous wounds, regenerative medicine

## Abstract

This study aimed to explore an alternative solution for treating chronic wounds in dogs that did not respond to standard treatments. The authors evaluate the application of autologous leukocyte- and platelet-rich fibrin membranes. These membranes were applied directly to the wounds, and healing was monitored over time. The results showed significant improvements in wound closure, tissue regeneration, and overall healing rates. Most wounds healed completely within three weeks. Importantly, no antibiotics were needed throughout the healing process, reducing the risk of antibiotic resistance and cutting costs. The use of these membranes not only accelerated recovery but also minimized the need for repeated interventions. This approach offers a promising, natural alternative for managing chronic wounds in dogs, with potential applications in veterinary and even human medicine. It demonstrates how innovative therapies can improve health outcomes by enhancing conventional treatments.

## 1. Introduction

Wound healing, the innate restorative response to tissue injury, involves a cascade of complex, orderly, and elaborate events involving many cell types and is guided by the release of soluble mediators and signals, which can influence the homing of circulating cells to damaged tissue [1,2]. However, local and systemic factors can disrupt these processes and delay healing [3]. Local wound environment factors include pressure, perfusion, infection, persistent foreign material, tissue viability, hematoma, and/or seroma. Systemic factors, conversely, include endocrine disorders, oncologic or autoimmune diseases, hypertension, obesity, malnutrition, and peripheral vascular disease [3,4]. Wounds that fail to heal despite appropriate treatment and are arrested in a prolonged and dysregulated inflammatory to proliferative phase leading to a persistent non-healing state are defined as chronic wounds [4,5,6].

Research on the relationship between wound healing and molecular biology has revealed that the normal healing process depends highly on growth factors and cytokines [7]. Platelets are important cells that regulate the hemostasis phase through vascular obliteration and facilitate fibrin clot formation [2], but they also play an essential role in the healing process. Platelet concentrates have been used for more than four decades for their positive effects on tissue regeneration by promoting angiogenesis and cell recruitment, proliferation, remodeling, and differentiation [8]. Platelet-rich fibrin (PRF) is a second-generation platelet concentrate obtained by centrifugation of autologous peripheral blood [9]. PRF can be divided into pure-PRF and leukocyte- and platelet-rich fibrin (L-PRF) preparations, and new centrifugation protocols are being developed to prepare new platelet derivatives (A-PRF, i-PRF, i-PRF M, i-PRF +) [10].

L-PRF is characterized by the high concentration of platelets, which can release growth factors (GFs), such as platelet-derived growth factor (PDGF-AA, PDGF-AB, PDGF-BB), transforming growth factor-β1 (TGF-β1), vascular endothelial growth factor (VEGF), and other important proteins in the surrounding tissue [10]. Moreover, unlike platelet-rich plasma (PRP), L-PRF also contains an elevated concentration of white blood cells, which have a crucial role in wound healing [11]. White blood cells, including neutrophils and macrophages, are among the first cell types recruited into injured tissues, and their function includes phagocytosis of debris, microbes, and necrotic tissue to prevent infection. Macrophages are also key myeloid cells, which are essential for their ability to secrete growth factors, such as TGF-β1, PDGF, and VEGF [11,12].

Despite the growing number of scientific studies investigating the application of platelet concentrates in humans [13,14,15], there are only a few papers [16,17,18] regarding using L-PRF for wound healing in dogs and cats. The veterinary literature has mainly focused on using platelet derivatives in experimentally acute induced cutaneous wounds and exclusively on using PRP [19,20,21,22,23]. Nevertheless, the treatment of chronic cutaneous wounds remains an important challenge in both human and veterinary medicine. The aim of this prospective study was to assess the clinical efficacy of L-PRF membranes obtained by a standardized production protocol [11] for treating chronic wounds in dogs. We hypothesized that L-PRF could be a cost-effective and easy-to-manufacture product to promote wound healing in challenging clinical scenarios in dogs.

## 2. Materials and Methods

### 2.1. Inclusion Criteria

The study population included dogs referred to the Veterinary Teaching Hospital and the Department of Veterinary Medicine and Animal Production of the University of Naples “Federico II” between June 2022 and June 2023 diagnosed with chronic cutaneous wounds and treated with L-PRF membranes. A chronic cutaneous wound was defined as a full-thickness cutaneous defect that failed to heal within 4–6 weeks despite appropriate treatment, regardless of the location or etiology [6]. Animals for which secondary wound closure was performed, animals whose owners declined to be part of the study, animals lacking adequate follow-up information, and animals diagnosed with thrombocytopenic conditions or infectious or neoplastic disease were excluded from the study. Informed consent was obtained from the owners of enrolled dogs.

Collected data include animal breed, age, weight, sex, previous treatments, time of wound onset before L-PRF treatment, wound etiology, and affected body region.

### 2.2. Study Assessments

A complete physical examination, blood count, and serum biochemical analyses were performed on all dogs enrolled in this study.

At admission (T0), wounds were classified according to their anatomic location. In addition, digital photographs of the wounds were taken alongside a standardized ruler for digital calibration. Each captured wound image was imported into computer software (Adobe Photoshop 2022, version 25.2.0, Adobe Inc., San Jose, CA, USA), and the total wound area (TWA) was calculated in centimeters squared by tracing the wound edges (Figure 1).

The ruler was used as a size reference for image processing to quantify the wound area based on the pixel count. The TWA was measured three times, and the mean area in each animal was reported. The maximum length and width of the wounds were measured in centimeters. The epithelialization area (EA) and granulation tissue area (GTA) were measured in centimeters squared with pixel count. From the day of admission and the start of L-PRF treatment (T0), the percentage of epithelialization was determined using the following formula:$$\%of\;epithelialization=EA\;at\;T0\;{cm}^{2}/TWA\;at\;T0\;{cm}^{2}\times 100$$

At the time of admission, wounds were also independently assessed by two authors (FA, CC) using the Bates-Jensen Wound Assessment Tool (BWAT) [24], which includes four parameters (size, depth, edges, and undermining) scored on a range of 0–5, and nine parameters (necrotic tissue type and amount, exudate type and amount, color of the surrounding skin, peripheral tissue edema and induration, granulation tissue, and epithelialization) scored on a range of 1–5, yielding a minimum possible score of 9 and a maximum possible score of 65.

### 2.3. Treatments

After aseptic preparation of the affected area, an experienced surgeon (GDV) selected the appropriate treatment depending on the wound’s size, shape, environment, and location. The wound was treated with low-pressure irrigation with sterile lactated Ringer’s solution (1 L bag minimum), which allowed mechanical removal of particles and gross contaminations from the wound bed. If necrotic tissue was present (BWAT score for the “amount of necrotic tissue” of ≥2), surgical selective debridement was performed, and the TWA was calculated from postoperative digital photographs.

Each wound was treated with one or more L-PRF membranes, depending on the animal’s body weight and the shape and size of the wound, to ensure that the entire wound surface was covered with L-PRF membrane. For dogs weighing less than 8 kg, 5 mL of blood was collected, and a single membrane was applied to wounds with a TWA of less than 10 cm^2^. The L-PRF membranes were obtained as described by Caterino et al. [11]. Briefly, 9 mL of autologous blood sample was collected from the jugular vein of each dog and centrifuged within 2 min using a fixed angle desk centrifuge (TD4A-WS, In LoveArts, Qingdao, China) with a sequential protocoled speed. After centrifugation, the fibrin clot (L-PRF clot) was separated from the platelet-free plasma on the top and the red clot at the bottom. The fibrin clot was then placed in the L-PRF Wound Box, where a homogeneous compression was applied for 15 min, obtaining the L-PRF membrane. Before the membrane application, the fluid extracted from the L-PRF clot under compression was used to irrigate the wound bed using a sterile syringe. The L-PRF membranes were then sutured to the wound edges by a simple interrupted suture pattern using nonabsorbable monofilament suture (4.0–2.0 Ethilon, Ethicon, Amersfoort, the Netherlands), trying to cover the entire surface of the wound bed (Figure 2).

A sterile nonadherent dressing pad (Telfa, Covidien, Mansfield, MA, USA) was used as the first layer of a wound dressing, followed by a layer of cotton gauze and a tertiary layer of self-adherent dressing wrap (Vetrap, 3M Health Care, St. Paul, MN, USA). No topical agents/ointments were used for wound dressing. The bandages were changed every 2–3 days as needed until wound healing was complete. Elizabethan collars were used to prevent animals from interfering with the dressings and the healing process. Anti-edema and non-steroidal anti-inflammatory drugs were only used if the BWAT scores for exudate, peripheral edema, necrotic, or granulation tissue were >4. The administration of antibiotics was limited to wounds that showed signs of infection, such as severe soft tissue swelling, erythema, pain, hyperthermia, and/or purulent discharge.

Wound assessments were scheduled every week, but an earlier assessment was performed if necrotic tissue (BWAT score for necrotic tissue > 2) or exudation (BWAT score for exudate > 2) were present. Digital photographs of the wounds were taken during the follow-up to assess TWA, EA, and GTA (Figure 3) at 7, 14, 21, 30, and 45 days (T1, T2, T3, T4, and T5, respectively). Wound contraction, epithelialization, and healing were calculated using the following formulas:$$\%of\;wound\;contraction\;at\;day\;\left(x\right)=100-\left(\frac{TWA\;at\;T(x)\;{cm}^{2}}{TWA\;at\;T0\;{cm}^{2}}\times 100\right)$$
where the formula in the parentheses corresponds to the percentage of wound size on day (*x*).
$$\%of\;epithelialization\;at\;day\;(x)=\frac{EA\;at\;T(x)\;{cm}^{2}}{EA\;at\;T0\;{cm}^{2}}\times 100$$


$$\%of\;wound\;healing\;at\;day\;\left(x\right)=100-\left(\frac{GTA\;at\;T(x)\;{cm}^{2}}{TWA\;at\;T0\;{cm}^{2}}\times 100\right)$$


The formula in the parentheses corresponds to the percentage of the nonhealed area compared to the total wound area at T0.

Also, the BWAT was assessed at each wound assessment until the wound was entirely covered by epithelial tissue. Complications were recorded and graded as catastrophic, major, or minor according to the guidelines described by Cook et al. [25] throughout the study period.

Finally, the time to complete wound healing was recorded for each wound, and it was defined as the time between the start of L-PRF treatment and the complete wound re-epithelialization.

### 2.4. Data Analysis

The convenience sample was described by variables, such as age and weight, which are expressed as the mean ± SD.

Data were recorded on a spreadsheet (Microsoft Excel 2019. Microsoft Corporation, Redmond, WA, USA) and imported into a software package (IBM SPSS Statistics, Version 28.0; IBM Corporation, Armonk, NY, USA) for statistical analyses. After verifying the data were not normally distributed using the Shapiro-Wilk test, Friedman’s analysis of variance (ANOVA) test for related samples was used to assess whether there were statistically significant changes in BWAT, TWA, GTA, and EA at different time intervals. Pairwise multiple comparisons were performed as post hoc tests and reported as the percentage change in the mean rank at each time interval.

Furthermore, a nonlinear regression model was performed using the nls function [26,27] in the STATS package implemented in R statistical software (version 4.3.2) [28]. The significance level for all statistical tests was set a priori at *p* ≤ 0.05.

## 3. Results

Twenty-four dogs were referred for cutaneous chronic wounds unresponsive to previous treatments. Of these, six were excluded due to lacking information at follow-up (n = 3), thrombocytopenia due to Ehrlichia infection (n = 1), or the decision to perform secondary closure (n = 2). Therefore, only 18 dogs met the inclusion criteria. The mean ± SD age and body weight of the study population were 4.6 (±3.2) years and 11.4 (±6.9) kg, respectively. Twelve dogs were male, of which two were neutered, and six were female (Table 1). All animals had received unsuccessful conventional treatments before admission, including ointments, antibiotics, anti-inflammatories, and different types of wound dressings.

The complete blood count and serum biochemical analysis, performed at admission, were unremarkable in all dogs. The onset of skin lesions was known in 15 dogs, with a median of 30 days. All dogs had a history of trauma, resulting in delayed-healing wounds in 15 and fistula in 3 cases. The etiology was known in only 10 dogs. Three delayed-healing wounds resulted from bites, one from dehiscence of a surgical suture, one from Cushing’s syndrome, one from a collar-provoked lesion, and one from fulguration. One dog suffered from perineal fistulae, and two dogs had a foreign body fistula. For eight dogs, no information was available regarding the origin of the lesion, but trauma due to a car accident was suspected.

All dogs were referred for a second opinion due to the lack of healing after previous treatments. The skin injuries were distributed fairly evenly over the limbs (n = 12), trunk (n = 4), neck (n = 1), and head (n = 1). The median total wound area (TWA) at T0 was 9.1 (5.9–21.1) cm^2^. According to the software measurements used to calculate TWA, the area of the wounds ranged from 1.66 to 51.2 cm^2^. The granulation tissue area (GTA) accounted for 7.8 (3.4–9.8) cm^2^, and the epithelialization area (EA) was 2.5 (0.9–3.2) cm^2^. The maximum length and width were 4.2 (1.9–17.4) cm and 5.1 (1.0–14.5) cm, respectively. The presentation’s median Bates–Jensen Wound Assessment Tool score (BWAT) was 34.5 (31.5–40.0).

All but one of the dogs were treated with a single application of L-PRF, and only one membrane was applied to eight of the wounds. Two treatments with L-PRF, for a total of four membranes, were required for Case 13, which had a “pocket wound”. Two membranes to cover the entire wound surface were used in nine wounds. None of the animals required antibiotics. Eight dogs received anti-edema drugs and non-steroidal anti-inflammatory drugs after treatment with L-PRF membranes.

Follow-up, involving physical examination and photographic assessment, was continued for a median of 45 (55–77) days. The results of Friedman’s ANOVA test for related samples revealed significant differences in BWAT (*p* < 0.001), TWA (*p* < 0.001), GTA (*p* < 0.001), EA (*p* = 0.007), and contraction (*p* < 0.001) at different time intervals (Figure 4).

The median BWAT score was 25.0 (20.0–29.2) at T1, with percentage decreases of 42.2%, 55.0%, and 70.6% at T3, T4, and T5, respectively (*p* < 0.001). The BWAT score was 19.5 (17.1–23.2) at T2, 15.0 (14.7–19.5) at T3, 13.0 (13.0–18.0) at T4, and 13.0 (13.0–13.5) at T5 (Figure 5). These scores were significantly lower than the score at T0 (*p* < 0.001), with percentage decreases of 35.1%, 51.8%, 70.8%, and 75.5%, respectively.

The median TWA decreased steadily over time, from 9.1 (5.9–21.1) cm^2^ at T0 to 3.3 (1.2–5.0) cm^2^ at T2 (*p* = 0.002), to 0.5 (0.0–2.2) cm^2^ at T3 (*p* < 0.001), to 0.0 (0.0–1.8) cm^2^ at T4 (*p* < 0.001), and to 0.0 (0.0) cm^2^ at T5 (*p* < 0.001), with percentage decreases of 34.9%, 55.6%, 66.0%, and 73.5%, respectively.

The median GTA was 7.8 (3.4–9.8) cm^2^ at T0, 4.8 (1.7–5.5) cm^2^ at T1, 1.2 (0.8–3.1) cm^2^ at T2, 0.0 (0.0–1.7) cm^2^ at T3, 0.0 (0.0–1.6) cm^2^ at T4, and 0.0 (0.0) cm^2^ at T5. Analyzing the data, the scores at T3 (*p* < 0.001), T4 (*p* < 0.001), and T5 (*p* < 0.001) were different from those at T0, with percentage decreases of 57.3%, 65.9%, and 70.8%, respectively.

The median EA was 2.5 (0.9–3.2) cm^2^ at T0, 1.4 (0.8–2.1) cm^2^ at T1, 0.5 (0.0–0.7) cm^2^ at T2, 0.0 (0.0–0.9) cm^2^ at T3, 0.0 (0.0–0.6) cm^2^ at T4, and 0.0 (0.0) cm^2^ at T5. Between the presentation and the first follow-up, EA increased in half of the cases and decreased in half of the cases. Between T0 and T3, we found a significant reduction of 42.3% (*p* = 0.002) and percentage decreases of 50.1% and 52.3% at T4 (*p* = 0.006) and T5 (*p* = 0.004), respectively.

At the first time point after treatment, the median of wound contraction was 40.3% (23.9%–42.8%) at T1, 72.5% (58.1%–95.5%) at T2, and 88.9% (78.0%–97.4%) at T3. The median (IQR) contraction was 97.2% (80.6%–100.0%) at T4 and 100.0% (100.0%) at T5, with significant percentage increases of 68.0% (*p* = 0.007) and 35.0% (*p* = 0.002), respectively, compared with T1.

During the study period, the median percent change in wound healing was 62.0% (46.6%–84.4%) at T1, 87.4% (68.1%–98.8%) at T2, and 100.0% (90.7%–100.0%) at T3 (Figure 6). Compared with T1, the increases were significant at T3 (*p* = 0.008), T4 (*p* = 0.002), and T5 (*p* < 0.001), with significant percentage increases of 206.0%, 241.9%, and 256.4%, respectively. All wounds healed within 22 days. No side effects were noted, and no complications occurred.

Considering the variability of healing times, a nonlinear regression model was applied to estimate a common trend in healing lesions across the sample (represented by the dashed line in Figure 7).

## 4. Discussion

In this prospective clinical study, we evaluated the effects of L-PRF membranes on wound healing of 18 non-experimental full-thickness cutaneous chronic wounds. The present study showed that L-PRF membranes are a cost-effective and easy-to-manufacture product to promote chronic wound healing. In addition, the use of L-PRF in this study reflects a common clinical setting, where it could be applied to wounds of varying size and characteristics, unlike in experimental studies where the wound size and depth would be homogeneous in all animals.

According to our data, the first significant improvement in the healing process was detected 14 days after the application of the L-PRF membrane with a progressive favorable progression until the complete wound re-epithelization was achieved. These results reflect the trend of L-PRF to release growth factors. As Caterino et al. reported, this bioscaffold is characterized by extensive release of growth factors within 1 week [11]. Therefore, the healing process is clinically apparent within 2 weeks and progresses until completion. The ability to progressively release growth factors and cytokines over a longer period allowed us to attempt a single application of L-PRF membranes for treating delayed-healing wounds, except in one case [11]. In our study, we achieved complete wound healing in all cases at a median time of 22 days. In contrast to the experience gained in this experiment, it was previously reported that a single application of platelet concentrates did not yield satisfactory clinical results [29]. Of note, the greatest reductions in TWA, GTA, and BWAT scores were achieved in the first 3 weeks, after which the healing rate slowed. Indeed, between T0 and T3, we observed significant percentage decreases in TWA, GTA, and BWAT that exceeded 50%. To date, there are no publications regarding the frequency of treatment. Nevertheless, the secretory capacity of the membranes and our findings may suggest that treatment is required every 2 weeks.

Because our sample mainly consisted of stray dogs, we were unaware of the etiology of the skin lesions in most cases. Furthermore, for most cases, the time of onset was reported by colleagues or set as the date of hospitalization and thus was probably underestimated. However, all lesions were objectively graded according to size, shape, and biological evolution. Because these were non-experimental wounds with delayed healing, all wounds were characterized by severe impairment of granulation tissue formation, wound contraction, and epithelialization. Tissue loss and bone exposure were also frequent (n = 7), especially in degloving wounds on distal limbs (n = 4). In these cases, the L-PRF membranes were used as a bioscaffold over the bone, and complete coverage with granulation tissue was achieved within 1 week. Perforation of bone is one of the recommended treatments to improve wound healing over exposed bone [29,30], but this treatment is more invasive and prone to complications than applying L-PRF membranes. Other methods of treating wounds with exposed bone include the use of topical agents [5,31], cancellous bone grafting [32], and porcine-derived small intestinal submucosa [33], but these increase the duration and cost of therapy.

Perianal fistulas were another type of wound that was treated. Considering the complex etiopathogenesis of this condition, Monti et al. combined PRP with prednisone, tacrolimus, and metronidazole and hypothesized that PRP may have shortened the healing time [34]. In our sample, the treated fistulas healed within 18 days, which is encouraging because we did not administer immunosuppressive drugs or antibiotics.

Most of the treated lesions were located on the limbs (n = 11) or in areas of the body exposed to movement (n = 2). The main complication described in the literature for these wounds is contracture, which results from the contraction of second-intention wound healing over the flexion surface [5,35,36]. In our experience, the use of L-PRF membranes seems to avoid excessive tension during scar tissue formation. The ability to suture a three-dimensional scaffold that fills the area of tissue loss helps to distribute the tension evenly over the entire wound length. Indeed, the range of motion of all joints affected by the skin lesions was not restricted once full epithelialization had occurred.

Steroids present significant challenges to the wound healing process. A high steroid level delays angiogenesis, decreases vascular permeability, reduces phagocytosis, and suppresses fibroblasts to inhibit collagen synthesis, thus decreasing the rate of epithelialization [37]. We also admitted a dog affected by Cushing’s syndrome (Case 20) with a forelimb skin lesion that was referred to us 1 year after the wound onset. A single application of L-PRF achieved complete wound healing within 21 days. The complete filling of the tissue defect, the constant release of growth factors, and the mechanical effect of the fibrin network probably outweigh the local side effects of steroids.

Because the wounds included in this study were not standardized, the wound area was measured by tracing the area, using a reliable planimetric method, and using the BWAT to qualitatively assess the characteristics of the wound during the follow-up. At admission, lesions were classified as contaminated or infected, with a median BWAT score of 34.5. Chronically infected wounds are usually characterized by unhealthy, pale, weak, and friable granulation tissue and display impaired contraction and epithelialization [38]. In addition, bacterial, granulocytic, and macrophage collagenases degrade collagen, reducing wound strength [35,37]. Prior studies have claimed that PRF exerts effective topical antimicrobial activity [39,40,41]. For this reason, even though most of the lesions were contaminated (n = 12), we did not administer antibiotics during treatment or apply local antimicrobial/antiseptic drugs, and no septic complications occurred. These results are encouraging, especially from the one-health perspective of reducing the amount of systemic antibiotic drugs administered to veterinary patients [42]. Prior studies of PRF-treated pet wounds [16,43,44] all involved concomitant administration of antibiotics and anti-inflammatory drugs. Only the study by Khalifa and colleagues in 2021, which compared the effects of PRF and PRP on secondary intention wound healing, did not clearly declare the use of antibiotics, but the wounds in that study were surgically created under sterile conditions [45]. However, we should acknowledge the possibility that because the animals in our study had been treated elsewhere, many had received antibiotic therapy before admission.

One of the difficulties in treating small dogs with platelet derivatives is the need to collect large amounts of blood. The preparation of PRP is challenging because 12.5–15 mL of blood is required to produce 2–4 mL of PRP, but PRF and L-PRF have been prepared using 4–5 mL of blood in dogs and cats [11,16,17,45]. In the present study, the number of membranes and the required blood volume were determined based on the animal’s TWA and body weight. Because the body weight of 5 of 18 dogs did not exceed 8 kg and the TWA at presentation did not exceed 10 cm^2^, we treated them with only one L-PRF membrane, which was obtained by centrifugation of 5 mL of blood. Therefore, in the study sample, the extent of the lesions did not require exceeding a blood volume that would surpass the safety range for the patient, while maintaining the limitation of the direct proportion between the wound size and the number of membranes.

Few clinical studies in veterinary medicine have evaluated the use of PRF for treating naturally occurring full-thickness wounds [16,17,46], whereas induced lesions have been episodically investigated [44,45]. Conversely, several studies have used PRP, but the results were controversial. Positive results were obtained in experimental wounds as well as in non-healing and delayed-healing wounds in dogs [19,20,22,23,47,48]. However, some contradictory results have been reported, and some studies reported no positive effects of PRP [17,49,50,51].

The exploratory intent of this study, the small sample size, and the difficulty of preparing a control group with overlapping characteristics of non-experimental wounds only allow us to extrapolate the observational data. However, we should emphasize that a control group without similar treatment is unethical, as explained in an earlier article [16]. Nevertheless, we believe that the time to complete wound healing achieved with L-PRF membranes for treating non-experimental delayed wounds in dogs is a promising result. We achieved satisfactory healing of unresponsive lesions to previous treatments in all dogs. The observed healing trend confirms the validity of using L-PRF membranes, even with a single application. Nevertheless, any comparison with other studies in which other dressings or therapies were used carries a high risk of bias due to spontaneous wounds’ specific nature and characteristics. Further studies in the future could evaluate the clinical use of L-PRF derivates, such as A-PRF+, which seem to be characterized by a looser structure with more interfiber space and more cells [8].

## 5. Conclusions

In conclusion, in our experience, L-PRF membranes are inexpensive and easy to manufacture. The three-dimensional network of this bioscaffold makes it simple to apply during surgery and functional as well. Considering the results of this preliminary study, we believe that L-PRF membranes can become a reliable tool for treating lesions characterized by delayed healing. However, further clinical studies are needed to validate these findings more conclusively.

## Figures and Tables

**Figure 1 animals-15-00112-f001:**
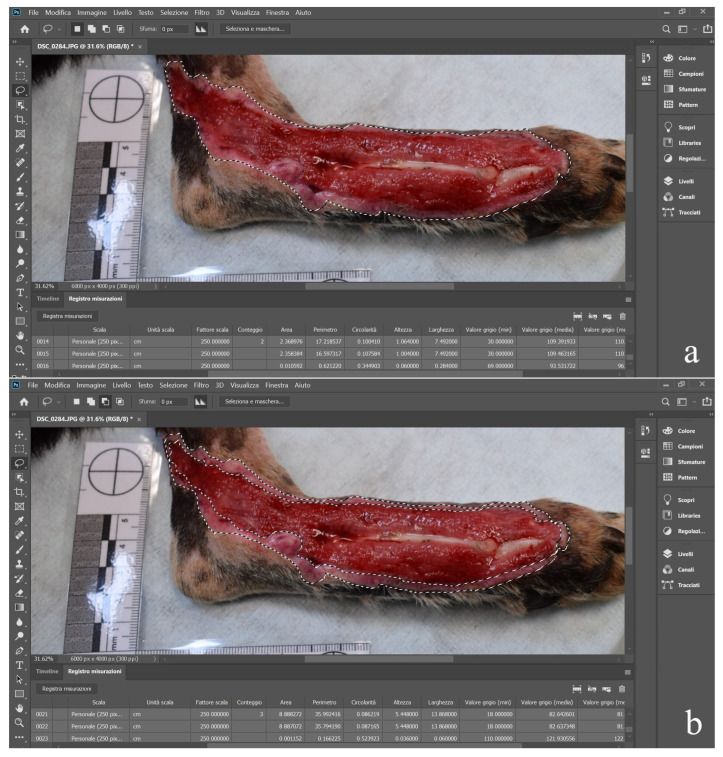
Total wound area at T0 (**a**) and epithelialization area at presentation (**b**) in Case 6.

**Figure 2 animals-15-00112-f002:**
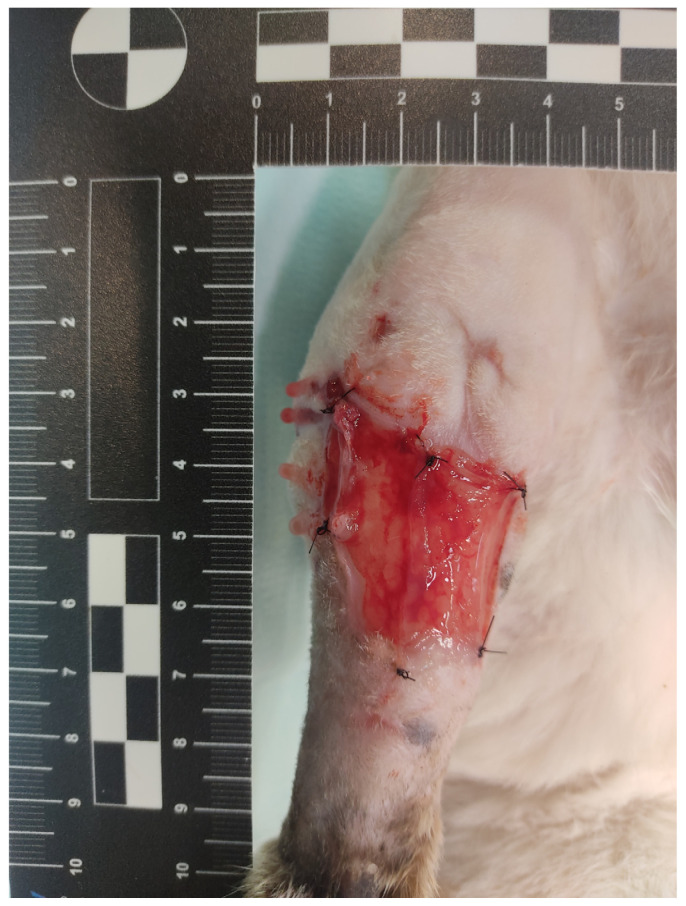
Case 15 after application of the L-PRF membrane.

**Figure 3 animals-15-00112-f003:**
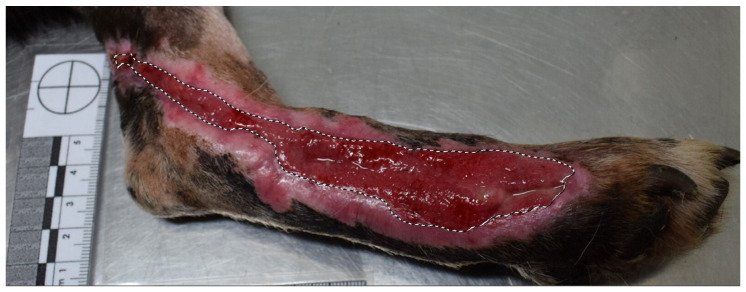
Granulation tissue area (GTA) at T1 in Case 6.

**Figure 4 animals-15-00112-f004:**
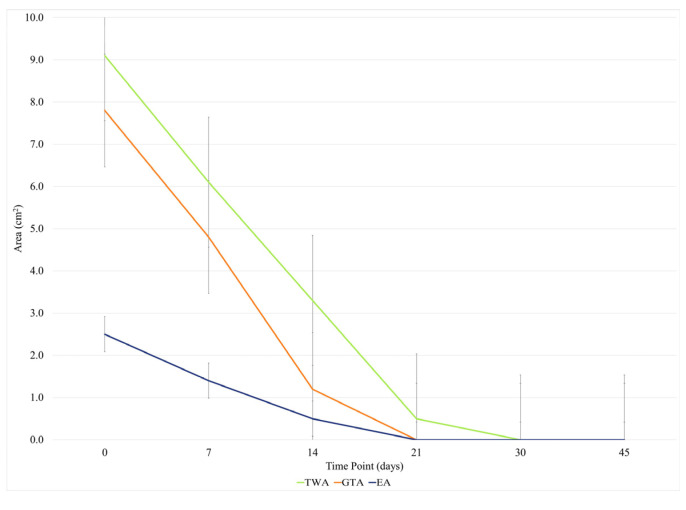
Total wound area (TWA), epithelialization area (EA), and granulation tissue area (GTA) during the follow-up. Values expressed as medians.

**Figure 5 animals-15-00112-f005:**
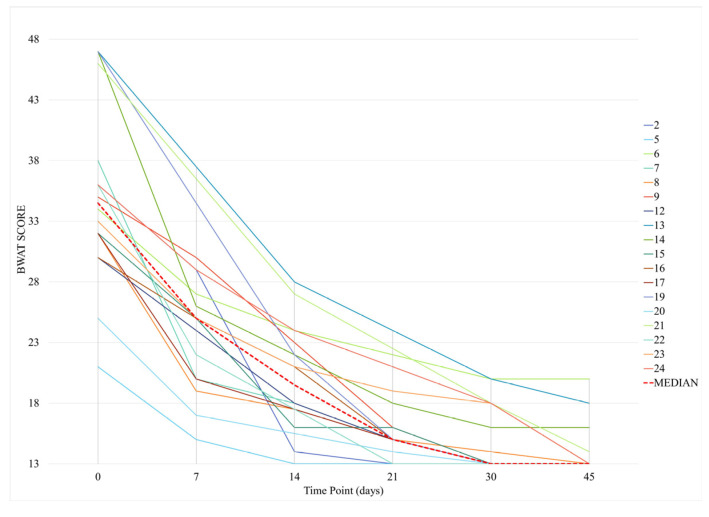
Bates-Jensen Wound Assessment Tool (BWAT) scores during the follow-up.

**Figure 6 animals-15-00112-f006:**
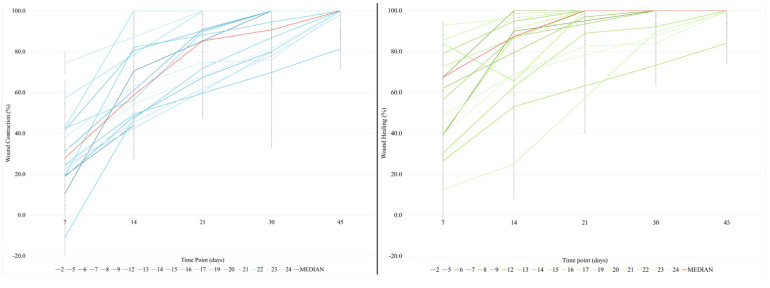
Wound contraction and wound healing during follow-up for all dogs.

**Figure 7 animals-15-00112-f007:**
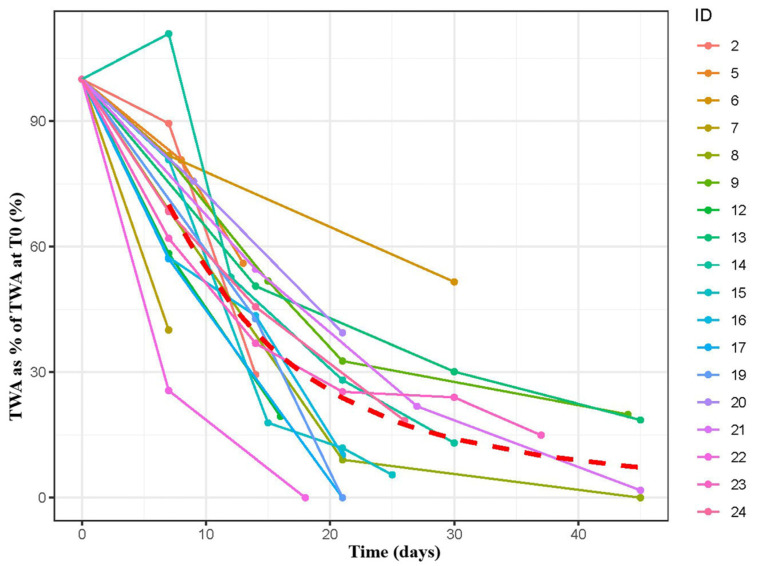
Change in total wound area (TWA) as a percentage of TWA at T0 in the individual dogs. The dashed line shows the nonlinear regression curve plotted using the nls function in R software.

**Table 1 animals-15-00112-t001:** Characteristics of the study sample: species, breed, age, sex, weight, onset, wound type, and location.

Case	Species	Breed	Age (Years)	Sex	Weight (kg)	Time Onset (Days)	Wound Location	Type of Wound	Origin
2	Dog	Mixed breed	5	M	15	Unknown	Forelimb	Delayed-healing wound	Unknown
5	Dog	French Hound	3	F	12	40	Flank	Fistula	Foreign body
6	Dog	Mixed breed	5	M	13	29	Tarsus and metatarsus	Delayed-healing wound	Unknown
7	Dog	Irish Setter	9	M	21	Unknown	Foot	Delayed-healing wound	Unknown
8	Dog	Volpino Italiano	0.5	F	2	7	Mouth	Delayed-healing wound	Fulguration
9	Dog	Miniature Pinscher	0.9	M	4	19	Homerus	Delayed-healing wound and open fracture	Bite
12	Dog	Mixed breed	3	F	12	25	Metatarsus	Delayed-healing wound	Unknown
13	Dog	Mixed breed	2	M	18	30	Left thoracic wall	Pocket Wound	Unknown
14	Dog	Segugio Maremmano	2	M	19	30	Metatarsus, phalanx, calcaneus	Delayed-healing wound	Unknown
15	Dog	Mixed breed	6	MC	5	28	Ulnar region	Delayed-healing wound	Unknown
16	Dog	Mixed breed	3	F	4	30	Perineal	Fistula	Unknown
17	Dog	Jack Russel	10	MC	7	45	Neck	Delayed-healing wound	Collar lesion
19	Dog	Chihuahua	1	M	5	5	Inguinal	Delayed-healing wound	Bite
20	Dog	Mixed breed	9	M	8	365	Radius	Delayed-healing wound	Cushing’s syndrome
21	Dog	Mixed breed	6	M	10	27	Forelimb	Delayed-healing wound	Bite
22	Dog	Mixed breed	7	M	22	Unknown	Perineal	Fistula	Perineal fistula
23	Dog	Mixed breed	0.75	F	3.5	30	Calcaneus	Delayed-healing wound	Unknown
24	Dog	English Setter	10	M	24	40	Tarsus	Delayed-healing wound	Dehiscence

## Data Availability

All data generated or analysed during this study are included in this published article.

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
