# Peer review of "Evaluation of the Effects of Autologous Leukocyte- and Platelet-Rich Fibrin Membranes for Treating Chronic Wounds: A Prospective Studyâ€"

_animals, 2025, doi:10.3390/ani15010112_

Round 1
Reviewer 1 Report
Comments and Suggestions for Authors
The authors have described preparing platelet-rich fibrin membrane from whole blood to treat wounds. This manuscript shows this is an effective and cheap option for wound treatment. One of the limitations of the study however, was that the authors could not leave an untreated control site for ethical reasons. Therefore, the efficacy should be compared to historical data. Moreover, a limitation of this method is that preparing L-PRF requires large amount of blood collection in case of larger wounds. L-PRF preparation needs to be covered in methods.
Author Response
The authors have described preparing platelet-rich fibrin membrane from whole blood to treat wounds. This manuscript shows this is an effective and cheap option for wound treatment. One of the limitations of the study however, was that the authors could not leave an untreated control site for ethical reasons. Therefore, the efficacy should be compared to historical data. Moreover, a limitation of this method is that preparing L-PRF requires large amount of blood collection in case of larger wounds. L-PRF preparation needs to be covered in methods.
Comment 1: One of the limitations of the study however, was that the authors could not leave an untreated control site for ethical reasons. Therefore, the efficacy should be compared to historical data.
Response 1: Thank you to pointing this out. As mentioned (line 403), it would have been unethical not to treat patients for the sole purpose of conducting a comparison. Moreover, because these were non-experimental wounds, a comparison would have been challenging to implement, even considering wounds previously treated without the use of PRF. Therefore, each subject, given that they were referred as a second opinion and had not responded to the initial "standard" treatment, was considered their own control.
Comment 2: a limitation of this method is that preparing L-PRF requires large amount of blood collection in case of larger wounds.
Response 2: Thank you for your comment. We add this aspect as a limitations of using of L-PRF membrane. See line 389-392: "Therefore, in the study sample, the extent of the lesions did not require exceeding a blood volume that would surpass the safety range for the patient, while maintaining the limitation of the direct proportion between the wound size and the number of membranes.”
Reviewer 2 Report
Comments and Suggestions for Authors
1) The numerical order of the figures is incorrect; Change it;
2) Did the PRF membranes cover the entire lesion or only a part? Specify;
3) It is not specified whether the centrifuge used to produce L-PRF is a fixed angle or a tilting one;
4) Why was compression applied to the Wound-Box for 15 minutes and not two as reported in the literature?; A two-minute compression does not release all the growth factors in the compression liquid;
5) On line 299, an unidentifiable Figure 2 is reported;
6) The Authors are advised to use the new form of PRF (A-PRF+) (1300 rpm x 8 min) in future experiments, which may require only one application compared to L-PRF;
Author Response
Comment 1: The numerical order of the figures is incorrect; Change it;
Response 1: Thank you for your comment. We revised and corrected the numerical order of the figures.
Comment 2: Did the PRF membranes cover the entire lesion or only a part? Specify;
Response 2: Thank you to pointing this out. The L-PRF membranes, single or multiple membranes, cover the entire lesion. We specified, accordingly to your observation, this point. Please, see line 140-141.
Comment 3: It is not specified whether the centrifuge used to produce L-PRF is a fixed angle or a tilting one;
Response 3: Thank you for your comment. We used a fixed angle centrifuge. We added, accordingly to your comment, this information to the manuscript. Please, see line 145.
Comment 4: Why was compression applied to the Wound-Box for 15 minutes and not two as reported in the literature?; A two-minute compression does not release all the growth factors in the compression liquid;
Response 4: Thank you to pointing this out. We followed the production protocol described by Caterino et al. (2022), therefore we applied a 15-minute compression. A compression duration of 15 minutes allowed for the formation of a membrane with sufficient size and thickness to be sutured; in any case, the fluid derived from the compression of the clot, rich in growth factors, is aspirated with a syringe and applied directly to the wound before the membrane is applied, in order to prevent loss of the fluid. Please see line 150-151.
Comment 5: On line 299, an unidentifiable Figure 2 is reported;
Response 5: Thank you for your observation. We removed “figure 2” at line 299-303.
Comment 6: The Authors are advised to use the new form of PRF (A-PRF+) (1300 rpm x 8 min) in future experiments, which may require only one application compared to L-PRF;
Response 6: Thank you for your observation. We added, accordingly to your comment, this information to the manuscript. Please, see line 410-412.
Reviewer 3 Report
Comments and Suggestions for Authors
Reviewer find the submitted work, addressing L-PRF therapy to be of high interest to the intended readership. Certainly, a wound healing treatment protocol that, may obviate the need for antimicrobial chemotherapy, particularly given the rise of resistant pathogens, constitutes great potential value. Likewise, an approach that induces consistently positive outcomes in the treated canine patient represents its own intrinsic reward.
In order to sufficiently improve the submitted work, a number of concerns observed by Reviewer must be addressed. To begin, throughout the manuscript, numerous grammatical errors are evident (these likely reflect difficulties arising during the process of translating into English the Author's native language).
For example, in Line 45 and elsewhere the word 'natural' is used somewhat inappropriately. Perhaps 'endogenous' or 'innate' might better serve as a replacement. Again, in line 95 the words 'fail to heal' should be modified to read 'failed to heal'. Another example may be found in line 217 where the words 'in only in' are printed. Reviewer requests Authors to ensure that the work is scrutinized and properly formatted for native English speaking readers.
In line 77, Authors state that "....Despite the growing number of scientific studies investigating the application of platelet 77 concentrates in humans, there are no data regarding the use of L-PRF for wound healing in dog and cat." Aside from the failure to properly use the words 'dogs and cats' Authors appear to have overreached a bit. Indeed, although few in number, there do seem to be published cites describing the use of L-PRF in the setting of canine wound healing: https://pubmed.ncbi.nlm.nih.gov/35498727/
Reviewer requests that the statement be rephrased to reflect the noted criticism.
Scrolling further down the document to line 299 Reviewer finds the following verbiage:
Figure 2. This is a figure. Schemes follow another format. If there are multiple panels, they should be listed as: (a) Description of what is contained in the first panel; (b) Description of what is contained in the second panel. Figures should be placed in the main text near to the first time they are 301 cited.
This appears to be an artifact, perhaps from an internal Review - i.e. prior to actual submission. It should be removed.
Between the paragraphs adjacent to lines 356 & 357, Authors seem to jump into a new topic (steroid issues) without giving Reader a chance to grasp that a newly expressed thought is occurring. Thus, Reviewer asks that Authors insert an appropriate bridge sentence here.
Reviewer finds the thought expressed beginning in line 384 to be perhaps one of the most important findings described in the paper. Authors should touch on this somewhere in the introduction section - as it constitutes a key reason why their proposed therapy modification approach offers highly salient value in a key canine patient cohort.
Finally, beginning in line 412 Authors overreach -- without noting the inherent limits in their work. Here, a thought such as 'to more conclusively validate, further clinical studies are needed - to be supported with basic science interrogation' needs to be incorporated.
Comments on the Quality of English Language
A thorough rewrite, with updates commensurate to the level of a native English language speaking reader, is required.
Author Response
Reviewer find the submitted work, addressing L-PRF therapy to be of high interest to the intended readership. Certainly, a wound healing treatment protocol that, may obviate the need for antimicrobial chemotherapy, particularly given the rise of resistant pathogens, constitutes great potential value. Likewise, an approach that induces consistently positive outcomes in the treated canine patient represents its own intrinsic reward.
In order to sufficiently improve the submitted work, a number of concerns observed by Reviewer must be addressed. To begin, throughout the manuscript, numerous grammatical errors are evident (these likely reflect difficulties arising during the process of translating into English the Author's native language).
Comment 1:For example, in Line 45 and elsewhere the word 'natural' is used somewhat inappropriately. Perhaps 'endogenous' or 'innate' might better serve as a replacement.
Response 1: Thank you for your observation. We changed with “innate”. Please see line 45.
Comment 2: Again, in line 95 the words 'fail to heal' should be modified to read 'failed to heal'.
Response 2: Thank you for your observation. We changed with failed to heal. Please see line 95.
Comment 3: Another example may be found in line 217 where the words 'in only in' are printed. Reviewer requests Authors to ensure that the work is scrutinized and properly formatted for native English speaking readers.
Response 3: Thank you for your observation. We removed “in”. Please see line 217.
Comment 4: In line 77, Authors state that "....Despite the growing number of scientific studies investigating the application of platelet 77 concentrates in humans, there are no data regarding the use of L-PRF for wound healing in dog and cat." Aside from the failure to properly use the words 'dogs and cats' Authors appear to have overreached a bit. Indeed, although few in number, there do seem to be published cites describing the use of L-PRF in the setting of canine wound healing: https://pubmed.ncbi.nlm.nih.gov/35498727/. Reviewer requests that the statement be rephrased to reflect the noted criticism.
Response 4: Thank you for your observation. The paper that you suggest is a description of a production protocol, not a clinical application for wound healing, therefore it was not been considered. Moreover, the the suggested paper was written by the same authors of this study. We added two other papers (already cited), please see line 78.
Comment 5: Scrolling further down the document to line 299 Reviewer finds the following verbiage:
Figure 2. This is a figure. Schemes follow another format. If there are multiple panels, they should be listed as: (a) Description of what is contained in the first panel; (b) Description of what is contained in the second panel. Figures should be placed in the main text near to the first time they are 301 cited.
This appears to be an artifact, perhaps from an internal Review - i.e. prior to actual submission. It should be removed.
Response 4: Thank you for your observation. We removed “figure 2” at line 299-303.
Comment 5: Between the paragraphs adjacent to lines 356 & 357, Authors seem to jump into a new topic (steroid issues) without giving Reader a chance to grasp that a newly expressed thought is occurring. Thus, Reviewer asks that Authors insert an appropriate bridge sentence here.
Response 5: Thank you for your observation. We added a bridge sentence “Steroids present significant challenges to the wound healing process”. Please see line 354.
Comment 6: Reviewer finds the thought expressed beginning in line 384 to be perhaps one of the most important findings described in the paper. Authors should touch on this somewhere in the introduction section - as it constitutes a key reason why their proposed therapy modification approach offers highly salient value in a key canine patient cohort.
Response 6: Thank you for your observation. We believe that this aspect diverges from the objective of our work. It was not our aim to compare PRP and L-PRF in terms of the blood volume to be collected.
Comment 7: Finally, beginning in line 412 Authors overreach -- without noting the inherent limits in their work. Here, a thought such as 'to more conclusively validate, further clinical studies are needed - to be supported with basic science interrogation' needs to be incorporated.
Response 7: Thank you for your observation. We added, accordingly to your comment, “further clinical studies are needed to validate these findings more conclusively.”. Please see line 418-419.
Round 2
Reviewer 3 Report
Comments and Suggestions for Authors
Reviewer thanks Authors for their conscientious efforts to refine the submitted manuscript -- updated modifications add clarity and utility.